# Image Vignetting Correction Using a Deformable Radial Polynomial Model

**DOI:** 10.3390/s23031157

**Published:** 2023-01-19

**Authors:** Artur Bal, Henryk Palus

**Affiliations:** 1Department of Data Science and Engineering, Silesian University of Technology, Akademicka 16, 44-100 Gliwice, Poland; 2Biotechnology Center, Silesian University of Technology, Bolesława Krzywoustego 8, 44-100 Gliwice, Poland

**Keywords:** image vignetting, image shading, vignetting correction, single-image vignetting correction, flat-field correction, vignetting modeling, approximation function, low-level vision

## Abstract

Image vignetting is one of the major radiometric errors that occur in lens-camera systems. In many applications, vignetting is an undesirable effect; therefore, when it is impossible to fully prevent its occurrence, it is necessary to use computational methods for its correction. In probably the most frequently used approach to the vignetting correction, that is, the flat-field correction, the use of appropriate vignetting models plays a pivotal role. The radial polynomial (RP) model is commonly used, but for its proper use, the actual vignetting of the analyzed lens-camera system has to be a radial function. However, this condition is not fulfilled by many systems. There exist more universal models of vignetting; however, these models are much more sophisticated than the RP model. In this article, we propose a new model of vignetting named the Deformable Radial Polynomial (DRP) model, which joins the simplicity of the RP model with the universality of more sophisticated models. The DRP model uses a simple distance transformation and minimization method to match the radial vignetting model to the non-radial vignetting of the analyzed lens-camera system. The real-data experiment confirms that the DRP model, in general, gives better (up 35% or 50%, depending on the measure used) results than the RP model.

## 1. Introduction

Image vignetting, also called image shading, is a phenomenon in which the image brightness is reduced from the optical center of this image toward its edges. The characteristic of vignetting depends on the parameters, mainly geometrical and optical, of the lens-camera system used and the parameters chosen during image acquisition, such as aperture size (f-number) and lens focal length (in the case of the usage of a varifocal or a zoom lens). Vignetting is usually unintentional and undesired. The appearance of this effect is particularly undesirable when there is a need for radiometric or quantitative image analysis, which is very common in different areas, e.g., astronomy [1,2]; microscopy [3,4,5,6]; and remote sensing applications using terrestrial [7,8], airborne [9,10,11,12,13] and spaceborne sensors [14,15], to name just a few of them. This phenomenon is also undesirable in the case of the use of computational imaging algorithms, such as the creation of high dynamic range (HDR) images [16,17], the stitching of static images to create panoramic [18,19,20] or mosaic images [3,4,5,6,21], as well as a panoramic real-time view [22]. Vignetting also affects the results of image analysis, including the results obtained using neural networks [23,24].

The best way to reduce vignetting is to remove its causes. This aim can be reached by, e.g., the usage of a lens with appropriate characteristics or setting appropriate exposure parameters. However, in practice, such actions are not always possible or may not produce the desired results. In such cases, hardware solutions can be supported by the usage of computational vignetting correction methods. Based on the causes of vignetting, there exist four main types of vignetting [25,26], that is:
Mechanical vignetting—the effect of this type of vignetting is the blockade of light rays on their way from a scene through a lens to a camera sensor; the result of this is a complete loss of information in certain areas of the image and a lack of data needed for computational vignetting correction methods.Optical vignetting—is related to the optical characteristic of the used lens, its characteristic can be change by change of the lens aperture size [27].Natural vignetting—refers to the loss of image brightness caused by a change of the viewing angle Θ for individual image pixels, it is modeled by the $\cos^{4}\left(\Theta \right)$ law [28].Pixel vignetting—is related to the geometrical size and optical design (in the case of the use of microlenses) of the image sensor of the camera [29].

The computational vignetting correction methods can be used to correct the last three of them; that is, they cannot be used to correct mechanical vignetting. What is more, there is not an easy way to correct each vignetting type independently; however, in practice there is no need to distinguish the causes of vignetting because all mentioned types of vignetting (apart from mechanical vignetting of course) are jointly corrected in a single procedure. It is also worth noting that vignetting correction, the parameters of which are very often calculated using reference images acquired from close distance, can be directly used to correct images obtained for distant objects—such a procedure is typical, e.g., for remote sensing applications. This advantage is due to the fact that the actual vignetting characteristics for fixed parameters of the lens-camera system used do not depend on the distance between the imaging system and an object, but on the viewing angle at which this object is visible from this system.

During the years of development, many different approaches to the vignetting correction problem have been presented in the literature. The vignetting correction methods can be roughly divided into two groups, that is, into the group of methods that do not use the reference image or images, and the group of methods that use such data. The methods which belong to the first group use, e.g.:
Physically-based models of vignetting [28,30]—the usage of these methods requires detailed knowledge about the parameters of the used lens-camera system, which is often unavailable; however, this approach is very useful during the design process of the optical system.A single image [31,32,33,34] or a sequence of images [18,19,25,26] of a natural scene or scenes to estimate the vignetting—in the case of these methods, the vignetting estimation is obtained as a result minimization of an objective function with the assumption that vignetting is a radial function, which limits the number of lens-camera systems for which these methods can be used. The effectiveness of these methods also depends strongly on many other factors, such as the precision of localization of corresponding pixels, uniformity of the analyzing scene, which limits their applicability. However, and this is a significant advantage, these methods can be used for already acquired images when the acquisition of new reference images is not possible; such situations are common in the case of, e.g., historical images.

To the second group belong, among others, probably the most frequently used methods, that is, those which are based on the idea of a “flat-field correction” approach.

In the “flat-field” approach, the vignetting is estimated based on a reference image of vignetting $I_{V}$, which presents a uniformly illuminated flat surface with a uniform color. This approach assumes that the main source of brightness differences in $I_{V}$ is the vignetting of the lens-camera system that was used; however, because the $I_{V}$ image is acquired by a real lens-camera system, this image also contains noise. The formation of the $I_{V}$ image can be described by the following formula
(1)$$I_{V}\left(i,j\right)=V\left(i,j\right)\;\cdot \;I_{flat}\left(i,j\right)+\varepsilon \left(i,j\right),$$
where *V* is a real vignetting of the analyzed lens-camera system and V(i,j)∈(0,1] (where 1 means no vignetting at pixel (i,j) and 0 means complete blocking of light), $I_{flat}$ is an ideal image of a scene with a reference flat surface, ε(i,j) is an image noise and (i,j) are, respectively, horizontal and vertical pixel coordinates. The vignetting estimate $\tilde{V}$ is established during the approximation process, denoted as approx(·), using the assumed vignetting model VM and the image $I_{V}$ as follows
(2)$$\tilde{V}\left(i,j\right)=\mathrm{approx}\left(VM,I_{V}\left(i,j\right)\right),$$
where $\tilde{V}\in \left(0,1\right]$. The corrected version of an acquired image *I*, that is, an image $\tilde{I}$, is obtained using the following formula
(3)$$\tilde{I}\left(i,j\right)=I\left(i,j\right)\;\cdot \;\tilde{V}{\left(i,j\right)}^{-1}.$$For the best correction results, the images $I_{V}$ and *I* should be acquired using the same lens-camera system, its parameters (e.g., focal length for zoom lenses), and exposure parameters (e.g., value of aperture f-number). It should also be mentioned that for the most accurate vignetting correction results, this procedure should be preceded by a dark frame correction, and the camera used for the acquisition of the $I_{V}$ and *I* images should have a linear characteristic or both images, $I_{V}$ and *I* are linearized, that is, the relationship between the R, G, B values of the (i,j) pixel and the luminous flux of light incoming to the corresponding pixels of the camera sensor is proportional.

Adjusting the flat-field method to the analyzed system is a matter of using the vignetting model, which is appropriate for the actual vignetting occurring in this system. In the literature, different parametric vignetting models have been presented, for example, the polynomial 2D (P2D) model [25,35,36], the exponential 2D polynomial model [35], the smooth non-iterative local polynomial (SNILP) model [37], the radial polynomial (RP) model [38], the hyperbolic cosine model [39], and the Gaussian function [40]. The last three mentioned vignetting models belong to a widely used approach, which assumes that the actual vignetting *V* of the lens-camera system has a cylindrical symmetry. This means that the vignetting V(i,j) can be modeled using a radial function $\tilde{V}\left(r\right)$, that is, a function of *r*, where
(4)$$r=\sqrt{{\left(i-x_{C}\right)}^{2}+{\left(j-y_{C}\right)}^{2}}$$
is the distance between the pixel with coordinates (i,j) and the optical center *C* of the lens-camera system, with coordinates $\left(x_{C},y_{C}\right)$.

The use of the assumption of radial vignetting simplifies the process of searching for the vignetting estimate because the approximation of the 2D function is replaced by a much simpler approximation of the 1D function. The consequence of using a simpler approximation function (vignetting model) is a reduction in the number of parameters needed to determine the vignetting estimate, for example, the P2D model needs $(s^{2}+3s+2)/2$, where the RP model needs only s+3 parameters; *s* is the degree of the approximation function used. The usage of radial vignetting is convenient and therefore very popular; however, because not every lens-camera system satisfies this assumption, this approach to vignetting correction is not universal and should be used carefully. The radial vignetting assumption is not satisfied by a large group of imaging systems, such as industrial lenses designed with reducing vignetting in mind, perspective-control and focal plane-control lenses (shift and tilt lenses), anamorphic lenses, etc.

Analyzing the information presented above, it can be concluded that in the literature there is a lack of model proposals that combine the simplicity of use and a small number of parameters, which are characteristic for the RP model, with the universality of, for example, the P2D model or the SNILP model. In the article, we present a novel model of vignetting, that is, the *Deformable Radial Polynomial* (DRP) model, which is our attempt to fill this gap. The idea of the new model is based on the observation that the vignetting in many lens-camera systems is not ideally radial, but is rather a radial vignetting, which is “squeezed” in one direction. The contribution of the DRP method is the use of a simple function for distance calculation, which is a slight modification of (Equation 4), to transform the non-radial vignetting observed in the image space into the radial vignetting in the distance space. This step allows, for the price of adding only one parameter describing the non-radiality of the vignetting, for the use of a radial vignetting model, e.g., a polynomial 1D model as in the case of the DRP model, for modeling actual non-radial vignetting of a real lens-camera system. To verify this idea, the vignetting correction results obtained from the DRP model were compared with the results obtained using the models RP and P2D, which are well-known from the literature, as well as the state-of-the-art model, that is, the SNILP model. In the comparison, the data from five webcams with different vignetting characteristics, including the degree of non-radiality of their vignetting, were used. The results obtained show that the DRP model, in general, gives better results than the RP model. In the same cases, these results are up 35% or 50%, depending on the measure used, better than the results obtained from the RP model.

The rest of the article is organized as follows. The DRP model of vignetting is described in Section 2. Section 3 presents the conditions of the real-data model comparison, as well as its results. The discussion of the results is given in Section 4. The last, Section 5, contains the conclusions and suggestions for future research.

## 2. The Deformable Radial Polynomial Model of Vignetting

As mentioned above, the concept of estimation of vignetting using the RP model is very simple: it is assumed that the vignetting has cylindrical symmetry. Therefore, when analyzing the vignetting problem in the image plane, the value of the vignetting has rotational symmetry with respect to the optical image center *C*. This means that the vignetting model also has rotational symmetry, which leads to the use of the radial function as a vignetting model. In such a case, the vignetting value of all pixels that are at the same distance *r* from *C* has the same value v(r), and hence the initial 2D problem of finding the vignetting estimate can be solved as a much simpler 1D approximation problem. Thus, when as the 1D approximation function the 1D polynomial function of degree *s*, denoted as ${\mathrm{PolyReg}}^{s}\left(\;\cdot \;\right)$, is chosen, the radial polynomial (RP) model is obtained, which can be formally written as follows
(5)$$\tilde{V}={\mathrm{PolyReg}}^{s}\left(v(r)\right).$$
To solve this problem, that is, to find the parameters of the approximation polynomial, assuming that v(r) does not contain outliers, the ordinary least squares (OLS) method can be used.

When analyzing the images of vignetting $I_{V}$ (Figure 1), it can be seen that vignetting often is not a really radial function, but rather is a “squeezed” radial function. From this observation the founding idea of the DRP model has been derived, which is that the usage of the distance function, which is used to calculate distance *r* between the pixel (i,j) and the optical center *C* with coordinates $\left(x_{C},y_{C}\right)$, to transform the non-radial vignetting observed in the image space into radial vignetting in the distance space *r*. This simple transformation allows the use of a radial vignetting function, which in the case of the DRP model is a 1D polynomial function, for modeling actual non-radial vignetting of a lens-camera system.

The method for a simple realization of non-radial to radial vignetting transformation is taken from the method of modeling non-radial vignetting presented in [37]. According to the solution presented there, to accomplish such transformation it is sufficient to slightly modify Equation (Equation 4) by a parameter η, as follows
(6)$$r_{\eta }=\sqrt{{\left(i-x_{c}\right)}^{2}+{\left(\eta (j-y_{c})\right)}^{2}}.$$
The η can be treated as the measure of non-radiality of the vignetting, where, for ideal radial vignetting η=1, and for non-radial vignetting η≠1, e.g., for η>1, the vignetting in the vertical (*y*) direction is stronger than in the horizontal (*x*) direction.

The value of the coefficient η must be estimated. This can be found by solving the following minimization problem
(7)$$\eta^{*}=\arg\min_{\eta }\left(\frac{1}{MN}\sum_{i=1}^{M}\sum_{j=1}^{N}{\left(v\left(r_{\eta }\left(i,j\right)\right)-{\mathrm{PolyReg}}^{s}(v\left(r_{\eta }\left(i,j\right)\right))\right)}^{2}\right)$$
for the given coordinates $\left(x_{c},y_{c}\right)$ of the optical center *C* of the image; as mentioned earlier, the parameters *p* of ${\mathrm{PolyReg}}^{s}\left(r_{\eta }\left(i,j\right)\right)$ can be effectively found using the OLS method. When the optimal value of $\eta^{*}$ is determined, $\tilde{V}$ is calculated using the parameters *p* obtained from the ${\mathrm{PolyReg}}^{s}\left(v\left(r_{\eta }\left(i,j\right)\right)\right)$ for $\eta^{*}$. Because the acquisition of $I_{V}$ should be carried out in such a way that there is no saturation phenomenon in this image, this means that $\max(I_{V})<1$; assuming that the values of the pixels are represented in the normalized range [0,1]. This leads to a situation where $\max(\tilde{V})<1$; hence, it is necessary to normalize the range of $\tilde{V}$ to the range (0,1], which is performed as follows
(8)$$\tilde{V}\left(i,j\right):=\frac{\tilde{V}\left(i,j\right)}{\max_{i,j}\left(\tilde{V}\left(i,j\right)\right)},$$
where := is the assignment operator. The obtained $\tilde{V}$ is the final result of the vignetting estimate using the DRP model.

In the described above algorithm, the coordinates of *C* must be known or at least estimated before the DRP model can be used. However, these coordinates can also be estimated by solving the following multivariate minimization problem
(9)$$\eta^{*},{x_{c}}^{*},{y_{c}}^{*}=\arg\min_{\eta ,x_{c},y_{c}}\left(\frac{1}{MN}\sum_{i=1}^{M}\sum_{j=1}^{N}{\left(v\left(r_{\eta ,x_{c},y_{c}}\left(i,j\right)\right)-{\mathrm{PolyReg}}^{s}(v\left(r_{\eta ,x_{c},y_{c}}\left(i,j\right)\right))\right)}^{2}\right).$$
In this case, the final value of $\tilde{V}$ is calculated analogously as in the previous variant of the DRP model, that is, using the parameters *p* of ${\mathrm{PolyReg}}^{s}\left(v\left(r_{\eta ,x_{c},y_{c}}\left(i,j\right)\right)\right)$ estimated for the obtained $\eta^{*},{x_{c}}^{*},{y_{c}}^{*}$ and then normalized according to (Equation 8).

## 3. Experimental Comparison of Vignetting Models

### 3.1. Assumptions of the Comparison and Methods of Evaluation

The purpose of the conducted experiment was a comparison, based on real data, of the DRP model with the selected, known from the literature, vignetting models, that is, the RP model and the P2D and the state of the art SNILP models. The quality of the vignetting models can be evaluated by analyzing the image correction results (10)$${\tilde{I}}_{flat}=I_{V}\;\cdot \;{\tilde{V}}^{-1}.$$ obtained using vignetting estimates $\tilde{V}$ calculated using the compared vignetting models.

Ideal correction results should be flat, that is, all pixels in the corrected image ${\tilde{I}}_{flat}$ should have the same value. Of course, in the case of real data, this ideal result cannot be achieved due to the presence of noise in the input image $I_{V}$. In such cases, the pixels in the ${\tilde{I}}_{flat}$ image should have similar values with a possible minimal dispersion. Good dispersion measures are the standard deviation (STD) and the interquartile range (IQR); therefore, these measures were used for a quantitative evaluation of the results of the vignetting correction. Hence, a lower value of $\mathrm{STD}\left({\tilde{I}}_{flat}\right)$ or $\mathrm{IQR}\left({\tilde{I}}_{flat}\right)$ means a better ability of the analyzed vignetting model to adapt to the real vignetting.

Since the $I_{V}$ image contains noise and the estimation result $\tilde{V}$ may differ significantly from the vignetting of a given camera, it is possible that the ${\tilde{I}}_{flat}$ image contains pixels with values beyond the range of [0,255], that is, the range of pixel values for cameras used in the experiment. Therefore, the result of the correction, that is, the image ${\tilde{I}}_{flat}$, before its evaluation, is subjected to the operation of truncation of the pixel values according to the following formula
(11)$${\tilde{I}}_{flat}\left(i,j\right):=\left\{\begin{array}{ccc} 0 & \mathrm{for} & {\tilde{I}}_{flat}\left(i,j\right)<0\\ {\tilde{I}}_{flat}\left(i,j\right) & \mathrm{for} & {\tilde{I}}_{flat}\left(i,j\right)\in \left[0,255\right]\\ 255 & \mathrm{for} & {\tilde{I}}_{flat}\left(i,j\right)>255 \end{array}.\right.$$

### 3.2. Laboratory Setup and Data Acquisition Process

The data analyzed in the experiment were acquired using together five webcams, namely Microsoft LifeCam Studio, Logitech C920, Hama C-600 Pro, and two Xiaomi IMILAB CMSXJ22A webcams (Figure 2), which are indicated in the article, respectively, from Cam-A to Cam-E. In the case of webcams, their manufacturers usually do not provide detailed technical data, such as lens focal length, lens speed, etc. Despite this, a comparison of the main technical data of the webcams used is presented in Table 1.

The webcams used in the comparison were selected to differ in their vignetting characteristics. Such a selection of cameras allows for performing a comparison for different examples of real vignetting with a different vignetting strength (also called vignetting magnitude) and a different degree of non-radiality of the vignetting function, which is essential in the case of this comparison. Vignetting strength can be measured using the STD and IQR measure for the images $I_{V}$ acquired using individual cameras—the values of these measures are presented in the second column of, respectively, Tables 3 and 4. The tested cameras are arranged in the order from the camera with the lowest vignetting strength (Cam-A) to the camera with the highest vignetting strength (Cam-E). In Table 5, the estimated values of parameter η are presented. For simplicity of results analysis, the η value estimated during calculation of the best DRP-2 model for individual camera is in the article used as coefficient of vignetting non-radiality of this camera, and denoted as $\tilde{\eta }$. If the cameras were to be ranked in the order from the camera with the most radial vignetting to the camera with the most non-radial vignetting, that is in the order of ascending $\tilde{\eta }$ values, the order would be as follows: Cam-E, Cam-D, Cam-A, Cam-B, Cam-C.

As a flat-field surface, which is needed to acquire the vignetting image $I_{V}$, a uniformly back-lighted milky poly(methyl methacrylate) (PMMP, “plexiglass”) panel of size 50cm×100cm and thickness 3 mm was used. As a light source, the NEC EA304WMI-BK graphic monitor was used that displays a white screen with the brightness set to its maximum, i.e., 350 lx. The plexiglass panel was carefully placed parallel to the surface of the monitor screen, the distance between the monitor and the panel was constant and equal to 15 cm. To position the camera parallel to the monitor screen each camera was positioned in such a way that the geometric distortions of the reproduction of the test image obtained from this camera were symmetric. The test image was displayed on a monitor that served as panel illumination (NEC EA304WMI-BK), and the acquired images were observed in real time on the second display Figure 3a. For the time to acquire the series of images $I_{V}$, the aforementioned plexiglass panel was inserted between the monitor and the camera used Figure 3b.

To reduce the noise presented in the captured images, in the experiment, the ${\bar{I}}_{V}$ image was used, obtained as the average of 100 originally captured images $I_{V_{e}}$. Since the experiment evaluates the availability of fitting of the compared vignetting models to the real vignetting, there is no need to calculate the vignetting estimate for each color channel, that is, $\bar{R}$, $\bar{G}$, and $\bar{B}$, of the ${\bar{I}}_{V}$ image. Therefore, in the experiment, for each camera, as the input image,
(12)$$I_{V}=0.2989\bar{R}+0.5870\bar{G}+0.1140\bar{B}$$
was used. The exposure parameters of each camera were automatically carried out for the first image $I_{V_{1}}$, and for the rest of the images, that is, $I_{V_{2}},\ldots ,I_{V_{100}}$, the same parameters were used. The $I_{V_{e}}$ images were acquired in a dark laboratory room so that no additional lights interfered with this process. Additionally, all signaling diodes with which the cameras are equipped, as well as all contrasting inscriptions on the camera housings, were covered with black opaque tape. In the case of cameras equipped with autofocus, that is, for Cam-A to Cam-C, their focus has been set to infinity.

It is important to note that due to the objective of the comparison, eventually small errors in, e.g., positioning of the monitor, the plexiglass panel, or the cameras, as well as a small non-uniformity of screen illumination, do not influence the qualitative evaluation of the experiment results. Of course, such errors can affect the quantitative results of the estimation; however, these errors do not change the judgment of the ability of the vignetting models tested to find the best approximation $\tilde{V}$ based on the input image $I_{V}$, and precisely this property of the vignetting models is evaluated in the comparison.

### 3.3. Compared Vignetting Models and Their Implementations

The entire experiment, that is, from the image acquisition, through all calculations, to data presentation, was carried out using the MATLAB R2021b software package with the Image Acquisition Toolbox and the Optimization Toolbox. In the experiment, four vignetting models have been compared, that is, the novel deformable radial polynomial (DRP) model; known from the literature the radial polynomial (RP) [38] and polynomial 2D (P2D) [25] models; and the model which in terms of the quality of the vignetting correction can be treated as a state-of-the-art solution, that is, the smooth non-iterative local polynomial (SNILP) model [37].

The RP and DRP models have been tested in two variants, that is, with the calculation of coordinates $\left(x_{c},y_{c}\right)$ of the optical center of the image *C* performed before the vignetting estimation procedure (these variants are denoted, respectively, RP-1 and DRP-1) and with the estimation of $\left(x_{c},y_{c}\right)$ integrated into the vignetting estimation process (these variants are hereafter denoted as RP-2 and DRP-2). The implementation of all variants tested for the RP and DRP models uses the MATLAB functions polyfit and polyval, respectively, to estimate the parameters of the approximation polynomial and calculate its values. In the implementation of the RP-2, DRP-1, and DRP-2 models, the MATLAB function fminunc, which is a MATLAB procedure for solving unconstrained nonlinear optimization problems, is used to find the optimal values of $\left(x_{c},y_{c}\right)$ in the case of the RP-2 and DRP-2 models, and the values of η in the case of both variants of the DRP model. The coordinates of the optical center needed for the RP-1 and DRP-1 models were determined by searching for the coordinates of the maximum value of the 2D polynomial approximation of degree s=2 of the $I_{V}$ image. In the case of the RP-2 model for estimating $\left(x_{c},y_{c}\right)$ the optimization procedure (Equation 9) with η set as a constant value 1 has been used. As the initial values for the fminunc function, the following values were chosen: $\eta_{0}=1$, $x_{C_{0}}=\frac{M}{2}$ and $y_{C_{0}}=\frac{N}{2}$, where M×N is the resolution of the analyzed image, which for all cameras used was the same, that is, 1920×1080. For the others parameters of the fminunc function their default values have been used. The other two models, that is, the P2D and SNILP models, have been implemented according to the information given in [37].

All models were compared for different degrees *s* of the approximation polynomials, that is, for s∈{2,…,10}. The tested range of the values *s* covers the value of the degree of approximation polynomial proposed in the literature, that is, from s=2, for tasks that do not require exact correction, and up to s=6 in the most demanding applications. In Table 2, the comparison of type and number of parameters required to specify the vignetting estimates determined using individual models is presented.

### 3.4. Results of the Experiment

Figure 4 presents vignetting estimates $\tilde{V}$ obtained from the RP-1, DRP-2, and P2D models. These results are presented using pseudocolor and isolines, which allows for an easier comparison of the obtained vignetting estimates. In addition, in Appendix A in Figure A1, Figure A2, Figure A3, Figure A4, Figure A5, Figure A6, Figure A7, Figure A8, Figure A9 and Figure A10, the acquired vignetting images $I_{V}$, vignetting estimates $\tilde{V}$, and corrected images ${\tilde{I}}_{flat}$ are presented in the form of 3D charts.

The numerical results of the experiment—that is, the values of the STD and IQR measures calculated for the images ${\tilde{I}}_{flat}$, which are the results of vignetting correction—are presented in Table 3 and Table 4. In Figure 5, a comparison of the best results of the vignetting correction obtained for each model is presented. In this comparison, the STD and IQR values mentioned above are related to the values of the respective measures of the vignetting strength of the input images $I_{V}$; these values are presented in the second columns of, respectively, Table 3 and Table 4. Table 5 presents the estimated values of the coefficient η obtained for the DRP-1 and DRP-2 models.

## 4. Discussion of the Results

### 4.1. Evaluation of the Models in Terms of the Accuracy of the Obtained Vignetting Estimates

The purpose of the performed experiment was a comparison of the vignetting correction results obtained from the DRP model with the results obtained from the models known from the literature. Due to the large differences in the vignetting characteristics of the individual cameras used in the experiment, both in terms of the vignetting strength and the degree of non-radiality, the obtained correction results have been analyzed from the perspective of the influence of these factors on the correction results.

Analyzing the influence of the vignetting strength on the results of the vignetting correction, measured both with the STD (Table 3) and IQR (Table 4) measures, it can be seen that both tested variants of the novel DRP model, i.e., DRP-1 and DRP-2, practically always fall between the results obtained from the RP model and those obtained from the P2D and SNILP models. Slight exceptions to this general rule apply when low degree polynomials are used (s≤3) or when the correction was made for the image from the camera with near-ideal radial vignetting, that is, Cam-E with $\tilde{\eta }=1.0068$, where ideal radial vignetting has η=1. The same relationship between the results obtained from individual models also applies to the relative evaluation of the vignetting correction (Figure 5).

Among the cameras used for the experiment, only the Cam-E camera has an almost ideal radial vignetting ($\tilde{\eta }=1.0068$), while for the other cameras, the image brightness in the vertical direction decreases faster than in the horizontal direction (radial vignetting is ”squeezed” in the vertical direction). The speed of brightness decreasing in the case of other cameras varies from about 6% (Cam-D, $\tilde{\eta }=1.0605$), through about 9% (Cam-A, $\tilde{\eta }=1.0901$) and about 15% (Cam-B, $\tilde{\eta }=1.1559$) to almost 30% (Cam-C, $\tilde{\eta }=1.2952$). Evaluating the vignetting correction results obtained for Cam-E, the results obtained from all models are comparable. However, it can be noticed that for small values of s≤5, the RP and DRP models achieve better results than both more universal models, i.e., P2D and SNILP. This advantage decreases with increasing *s* and finally for s≥8 the P2D and SNILP models give better correction results than the RP and DRP models. It is worth noting that even in such a favorable situation for the RP model (almost ideal radial vignetting) for most values of *s*, any variant of the RP model does not give better results than the corresponding variant of the DRP model. These results indicate that it is worth considering using the DRP model instead of the RP model, even if the vignetting is almost ideal radial.

Interestingly, in the case of the Cam-E camera and the STD measure, the variants RP-1 and DRP-1, i.e., those that use the previously calculated coordinates $\left(x_{C},y_{C}\right)$ of the center *C*, give better results than the RP-2 and DRP-2 variants, in which these coordinates are sought within one optimization problem. A similar situation also occurs in the case of the Cam-A and Cam-D cameras, but only in the case of the IQR measure. Such an effect seems to be counter-intuitive, but it should be noted that the applied optimization procedure does not guarantee to find a global minimum, and no studies have been conducted on the influence of the selection of parameters of the optimization method on its results.

For each camera, which has a noticeable non-radial vignetting, that is, from Cam-A to Cam-D, the results obtained for any variant of the DRP model and for any value of *s* are always better than the results obtained for any variant of the RP model. With the increase in the degree of non-radiality (increasing $\tilde{\eta }$), this difference grows. In the case of the Cam-C ($\tilde{\eta }=1.2952$), the use of the DRP model in relation to the RP model, when comparing the corresponding variants of both models, gives about 35% improvement in the correction quality when the STD measure is used, and over 50% improvement in the case of the use of the IQR measure.

The influence of the degree of non-radiality on the results of vignetting correction is particularly well visible if one looks at the graphs in Figure 5. For the camera with almost radial vignetting (Cam-E), each tested model corrects the vignetting of this camera to a similar extent. However, with the increase in the non-radiality of camera vignetting, the more noticeable is the predominance of models that allow the occurrence of non-radial vignetting over the RP model. Of course, the DRP model is inferior to the P2D and SNILP models in terms of the obtained quality, but these models are much more complex. The difference in model complexity can be seen by comparing the number of parameters necessary to save the vignetting estimate obtained from each of these models (Table 2).

### 4.2. Computation Time

Another analyzed issue is the computation time, which is required to determine the vignetting estimate using individual models. In the vast majority of cases, even relatively long calculation times of the vignetting estimate are not a serious problem. The reason for this is that there is rarely a need to derive a vignetting estimate. Such situations occur occasionally, for example, when the new lens is used or periodically (e.g., every month) in order to ensure the highest quality of vignetting correction of the acquired images. Moreover, even if vignetting needs to be estimated for many parameters of the analyzed lens-camera system (e.g., for many aperture size values and many focal lengths of the varifocal lens), the calculations can be carried out in the background without involving the human operator.

In Figure 6 the median of the computational times for all models tested and for polynomial degrees s∈{2,…,10} are presented. In the test, the acquired images $I_{V}$ with resolution 1920×1080 were used and 25 repetitions (5 repetitions for each of the $I_{V}$ images acquired using tested cameras) were performed for each model-*s* value combination. The measurements were carried out on a computer equipped with an AMD Ryzen 7 3800X processor, 16 GB RAM, and a M.2 PCIe NVMe SSD disk using MATLAB 2021b software. The results obtained show that the DRP-2 model requires the longest computation time. This model is followed by the RP-2 and DRP-1 models. This situation results from the use of the minimization procedure, which is much slower than the OLS method used many times for individual lines of the image, as in the case of the SNILP model, or once, but for a larger number of data, as in the case of the P2D model.

It is also worth noting that the calculation times will vary as the degree of the polynomial increases. This is because that increasing the value of *s* increases also the size of the data matrix, and thus the number of mathematical operations that must be performed.

## 5. Conclusions and Future Research

The intention of the authors during the development of the DRP model was to combine the simplicity of the RP model with the universality of more complex models such as P2D and SNILP. The tests carried out for cameras, from which the acquired images differ in both the amount of vignetting and the degree of fulfillment of the radial vignetting assumption, fully confirm that this assumed goal has been achieved. The DRP model uses only one more parameter than the RP model, but according to the results obtained, it can give up to 35% better correction results in terms of the STD measure and up to over 50% better correction results in terms of the IQR measure.

It should be noted that the results obtained from the DRP model are better than those obtained from the RP model for all tested cameras, including the camera with almost perfect radial vignetting (Cam-E, $\tilde{\eta }=1.0068$), i.e., in conditions for which the RP model was designed. This observation leads to the essential conclusion that from the perspective of the quality of the vignetting correction, the DRP model can be successfully used as the default vignetting model instead of the RP model. Such a statement results from the fact that the results obtained from the DRP model in the case of radial vignetting are not worse than the results obtained from the RP model, and in the case of non-radial vignetting, they are better or much better than those obtained from the existing model. The expected quality improvement of the vignetting correction results increases with the increase in the degree of non-radiality of the analyzed vignetting.

The use of a polynomial function as a radial vignetting model is one of the few approaches to radial vignetting correction proposed in the literature (e.g., [39,40]). An interesting direction of research would therefore be to check how the usage of the distance transform (Equation 6) used in the DRP model would increase the flexibility of other than polynomial radial vignetting models. An important part of these studies should be the comparison of minimization methods in terms of their usefulness in the process of estimating the model parameters, the coordinates of the optical center *C* and the value of the $\eta^{*}$ coefficient.

In the conducted research, relatively simple lens-camera systems in the form of webcams were used. An important observation resulting from their application is the noticing of the prevalence of non-radial vignetting among the imaging systems used. Only one of the five cameras used has a vignetting that can be considered as radial (CamE) and only the other two have less than 10% differences in their vignetting in the vertical and horizontal directions (Cam-C and Cam-D). The question then arises to what extent the non-radial vignetting is widespread among more complex camera systems and how the developed DRP model will work in such systems. The analysis of these issues will be the subject of further research.

An important contribution made by the development of the DRP model—apart from making the RP model more flexible at the price of adding only one, easy to determine parameter η, which represents the degree of non-radial of actual vignetting—is that it opens the possibility of using methods adapted to images with radial vignetting for its use in the case of images with non-radial vignetting. Good examples of such methods are methods based on the idea of single-image vignetting correction [31,32,33,34]. This is possible because the DRP model uses transformation (Equation 6) of distance calculation between a pixel and optical image center *C* to transform the non-radial vignetting that occurs in most lens camera systems into an ideal radial vignetting model. As part of future research, it is planned to implement this idea in practice.

## Figures and Tables

**Figure 1 sensors-23-01157-f001:**
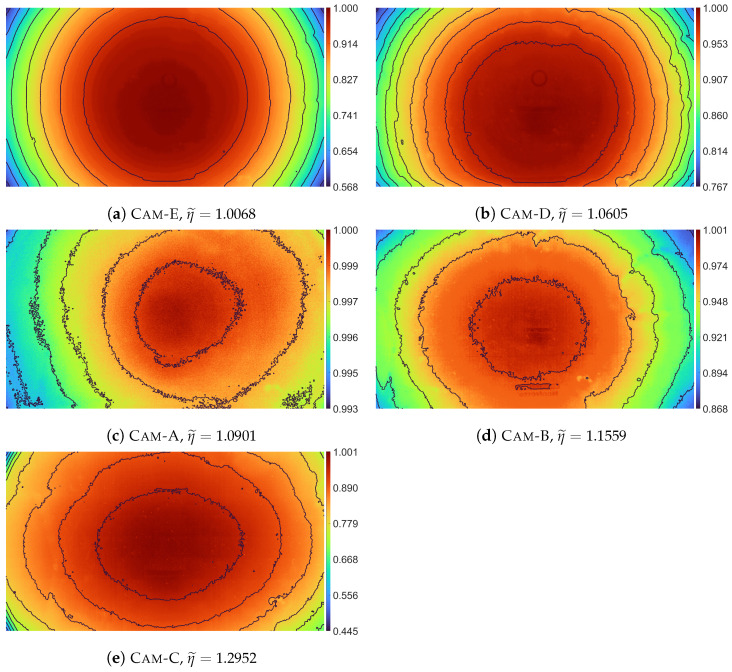
Comparison of the acquired images $I_{V}$; the images are presented in the order of the increasing non-radiality degree of vignetting, which is described by the $\tilde{\eta }$ coefficient.

**Figure 2 sensors-23-01157-f002:**
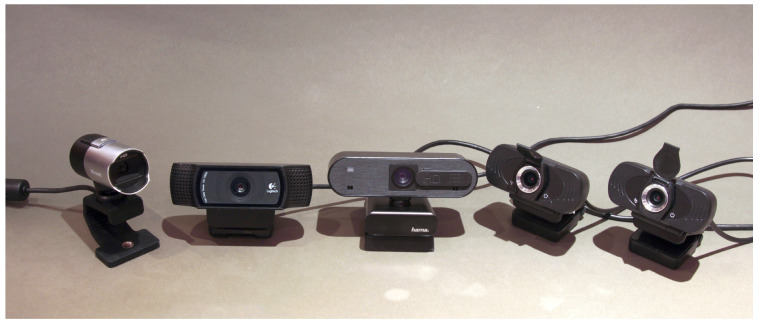
Cameras used in the comparison, from left to right: Microsoft LifeCam Studio (Cam-A); Logitech C920 (Cam-B); Hama C-600 Pro (Cam-C); and two Xiaomi IMILAB CMSXJ22A (Cam-D and Cam-E).

**Figure 3 sensors-23-01157-f003:**
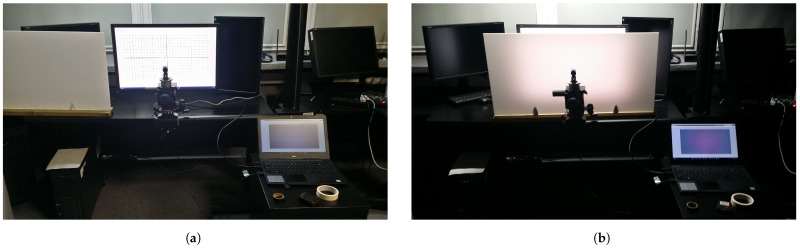
Laboratory setups used during: (**a**) positioning of camera; and (**b**) aquisition of the test images. The lighting in the laboratory was turned on only for taking the presented photographs.

**Figure 4 sensors-23-01157-f004:**
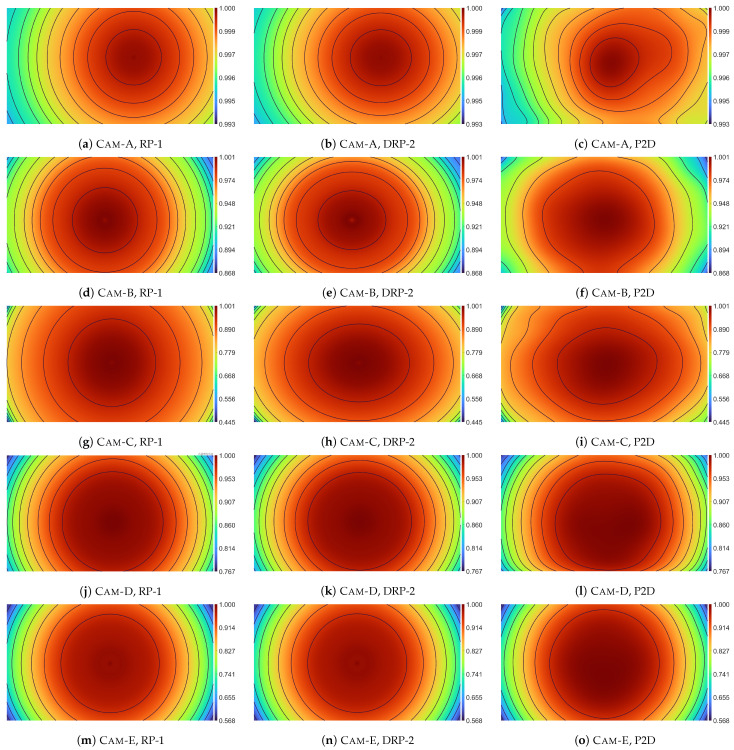
Comparison of the vignetting estimates $\tilde{V}$ obtained form the RP-1 (**left** column), DRP-2 (**central** column), and P2D (**right** column) models.

**Figure 5 sensors-23-01157-f005:**
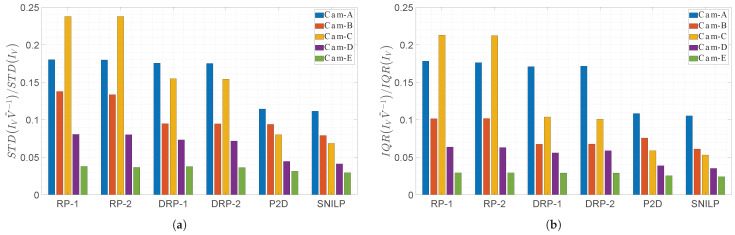
Comparison of the quality of the vignetting correction obtained for the compared vignetting models and used cameras. The values of measures of vignetting correction quality are presented in the relation to the initial values of the (**a**) STD and (**b**) IQR measures of the $I_{V}$ images obtained from the corresponding camera.

**Figure 6 sensors-23-01157-f006:**
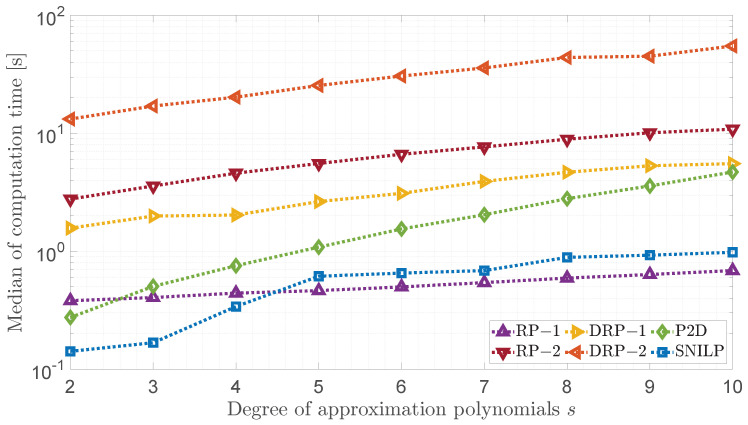
Comparison of vignetting estimation times for individual models.

**Table 1 sensors-23-01157-t001:** Main technical data of the cameras used in the comparison.

Parameters	Webcam			
	Cam-A	Cam-B	Cam-C	Cam-D & Cam-E
	Microsoft LifeCam Studio	Logitech C920	Hama C-600 Pro	Xiaomi IMILAB CMSXJ22A
Diagonal angle of view	75°	78°	90°	85°
Maximum video resolution	1920×1080	1920×1080	1920×1080	1920×1080
Maximal frame rate @ 1080p	30 fps	30 fps	30 fps	30 fps
Focus type	auto focus	auto focus	auto focus	fixed focus
Focus range	>10 cm	—	—	>60 cm

**Remark:** All data are provided by camera manufacturers.

**Table 2 sensors-23-01157-t002:** The type and the number of parameters required for saving vignetting estimation results.

Model	Type of Parameters	Number of Parameters
RP	parameters of 1D radial polynomial function + coordinates of image optical center $\left(x_{x},y_{c}\right)$	s+3
DRP	parameters of 1D radial polynomial function + coordinates of image optical center $\left(x_{x},y_{c}\right)$ + coefficient of non-radiality of the vignetting η	s+4
P2D	parameters of 2D approximation polynomial	$\frac{1}{2}\left(s^{2}+3s+2\right)$
SNILP	parameters of 1D approximation of each line along the longer side of the input image $I_{V}$ with the resolution M×N	N(s+1), where N≤M

**Table 3 sensors-23-01157-t003:** Comparison of $\mathrm{STD}\left(I_{V}{\tilde{V}}^{-1}\right)$ values.

Camera	$\mathrm{STD}\left(I_{V}\right)$	Model	Degree of Approximation Polynomials *s*								
			2	3	4	5	6	7	8	9	10
**Cam-A**	0.3028	RP-1	0.0553	0.0551	0.0550	0.0548	0.0548	0.0547	0.0545	0.0545	**0.0545**
		RP-2	0.0553	0.0550	0.0549	0.0547	0.0547	0.0546	0.0545	0.0545	**0.0544**
		DRP-1	0.0541	0.0538	0.0536	0.0533	0.0533	0.0533	**0.0531**	**0.0531**	**0.0531**
		DRP-2	0.0541	0.0537	0.0536	0.0533	0.0533	0.0532	0.0531	0.0531	**0.0530**
		P2D	0.0606	0.0585	0.0440	0.0429	0.0402	0.0398	0.0371	0.0368	**0.0346**
		SNILP	0.0540	0.0520	0.0410	0.0408	0.0377	0.0375	0.0349	0.0346	**0.0337**
**Cam-B**	3.7551	RP-1	0.5750	0.5749	0.5743	0.5634	0.5436	0.5379	0.5219	0.5189	**0.5165**
		RP-2	0.5630	0.5629	0.5628	0.5520	0.5331	0.5259	0.5068	0.5048	**0.5010**
		DRP-1	0.4549	0.4549	0.4543	0.4412	0.4170	0.4032	0.3681	0.3674	**0.3552**
		DRP-2	0.4549	0.4549	0.4542	0.4411	0.4167	0.4026	0.3676	0.3670	**0.3547**
		P2D	0.4533	0.4474	0.4201	0.4101	0.4005	0.3854	0.3732	0.3640	**0.3523**
		SNILP	0.4314	0.4219	0.4143	0.3970	0.3729	0.3553	0.3340	0.3128	**0.2962**
**Cam-C**	7.0446	RP-1	2.1372	2.0961	1.8347	1.7934	1.7297	1.7213	1.6779	1.6869	**1.6742**
		RP-2	2.1317	2.0870	1.8248	1.7840	1.7242	1.7185	1.6778	1.6866	**1.6740**
		DRP-1	1.4453	1.4042	1.2594	1.2135	1.1431	1.1207	1.0955	1.1008	**1.0894**
		DRP-2	1.4452	1.4016	1.2568	1.2120	1.1424	1.1190	1.0922	1.0974	**1.0852**
		P2D	1.4144	1.2838	1.2487	1.1791	0.9663	0.8157	0.7063	0.6197	**0.5633**
		SNILP	1.3209	1.1917	1.0207	0.9061	0.6955	0.6216	0.5582	0.5163	**0.4813**
**Cam-D**	9.9096	RP-1	0.9398	0.8891	0.8698	0.8339	0.8262	0.8014	0.7970	**0.7953**	0.7977
		RP-2	0.9308	0.8810	0.8641	0.8298	0.8217	0.7969	0.7927	**0.7909**	0.7933
		DRP-1	0.8353	0.7987	0.7945	0.7630	0.7554	0.7310	0.7252	**0.7221**	0.7243
		DRP-2	0.8267	0.7904	0.7860	0.7507	0.7434	0.7163	0.7105	**0.7066**	0.7087
		P2D	1.3315	1.3246	0.6429	0.6104	0.5923	0.5849	0.4956	0.4736	**0.4396**
		SNILP	1.1475	1.1377	0.6123	0.5962	0.5263	0.5035	0.4284	0.4188	**0.4079**
**Cam-E**	20.2225	RP-1	1.4188	1.1542	1.0754	0.8656	0.8570	0.8337	0.8349	0.8067	**0.7646**
		RP-2	1.4106	1.1428	1.0615	0.8393	0.8295	0.8074	0.8090	0.7811	**0.7364**
		DRP-1	1.3656	1.1200	1.0616	0.8582	0.8499	0.8247	0.8255	0.7998	**0.7600**
		DRP-2	1.3534	1.1047	1.0451	0.8294	0.8198	0.7956	0.7968	0.7717	**0.7297**
		P2D	2.8814	2.8686	1.0042	0.9923	0.9913	0.9857	0.6750	0.6726	**0.6339**
		SNILP	2.1547	2.1492	0.9763	0.9696	0.7148	0.7104	0.6072	0.6026	**0.5926**

**Remark:** (*i*) background colors (from blue, through light blue, light red to red) of Table cells represent range (respectively, from the best to the worst one) of the model results based on the STD(${\tilde{I}}_{flat}$) values obtained for each camera-*s* value combination; (*ii*) for each camera-model combination the values written with bold text represents the best approximation result obtained for the each of these combinations according to the STD(${\tilde{I}}_{flat}$) measure.

**Table 4 sensors-23-01157-t004:** Comparison of $\mathrm{IQR}\left(I_{V}{\tilde{V}}^{-1}\right)$ values.

Camera	$\mathrm{IQR}\left(I_{V}\right)$	Model	Degree of Approximation Polynomials *s*								
			2	3	4	5	6	7	8	9	10
**Cam-A**	0.3985	RP-1	0.0711	0.0706	**0.0704**	0.0708	0.0708	0.0711	0.0710	0.0710	0.0710
		RP-2	0.0704	0.0698	**0.0694**	0.0698	0.0698	0.0702	0.0702	0.0702	0.0702
		DRP-1	0.0687	0.0681	**0.0677**	0.0678	0.0678	0.0680	0.0679	0.0679	0.0680
		DRP-2	0.0690	0.0683	**0.0680**	0.0681	0.0681	0.0682	0.0682	0.0682	0.0683
		P2D	0.0749	0.0695	0.0572	0.0563	0.0522	0.0517	0.0456	0.0450	**0.0431**
		SNILP	0.0700	0.0678	0.0540	0.0536	0.0478	0.0474	0.0436	0.0432	**0.0419**
**Cam-B**	5.8670	RP-1	0.6895	0.6870	0.6887	0.7092	0.6474	0.6646	0.6088	0.6045	**0.5947**
		RP-2	0.6843	0.6814	0.6816	0.7046	0.6503	0.6672	0.6116	0.6075	**0.5952**
		DRP-1	0.5949	0.5949	0.5782	0.5891	0.5014	0.5157	0.4328	0.4302	**0.3953**
		DRP-2	0.5943	0.5945	0.5777	0.5882	0.4994	0.5138	0.4328	0.4303	**0.3961**
		P2D	0.6065	0.6030	0.5519	0.5341	0.5194	0.4996	0.4880	0.4803	**0.4437**
		SNILP	0.5670	0.5566	0.5524	0.5150	0.4817	0.4632	0.4199	0.3874	**0.3571**
**Cam-C**	10.1281	RP-1	2.4483	2.5710	2.2063	2.2129	2.2446	2.1988	**2.1458**	2.1781	2.1546
		RP-2	2.4640	2.6363	2.2521	2.2438	2.2693	2.2108	2.1490	2.1742	**2.1480**
		DRP-1	1.0902	1.1150	1.1001	1.1050	1.0742	1.0628	1.0471	1.0610	**1.0488**
		DRP-2	1.0903	1.1275	1.1127	1.1124	1.0709	1.0464	1.0261	1.0361	**1.0192**
		P2D	1.1020	0.9574	0.8716	0.8949	0.7357	0.6782	0.6468	0.6107	**0.5950**
		SNILP	1.0439	0.9015	0.7671	0.7554	0.6204	0.5927	0.5807	0.5536	**0.5346**
**Cam-D**	15.1372	RP-1	1.1225	1.1434	1.1373	0.9859	0.9875	**0.9488**	0.9515	0.9562	0.9616
		RP-2	1.0876	1.1134	1.1192	0.9662	0.9727	**0.9373**	0.9402	0.9454	0.9505
		DRP-1	1.0026	1.0831	1.0787	0.8987	0.8891	0.8373	**0.8358**	0.8427	0.8439
		DRP-2	1.0375	1.1085	1.1042	0.9478	0.9302	0.8815	**0.8785**	0.8847	0.8871
		P2D	1.6181	1.6353	0.9089	0.8706	0.8626	0.8409	0.6245	0.5941	**0.5837**
		SNILP	1.7324	1.7259	0.8850	0.8710	0.7243	0.6841	0.5549	0.5457	**0.5315**
**Cam-E**	31.6573	RP-1	1.1478	1.2411	1.2495	0.9811	0.9868	0.9932	0.9896	0.9831	**0.9224**
		RP-2	1.1565	1.2377	1.2452	0.9916	0.9915	1.0049	1.0004	0.9872	**0.9216**
		DRP-1	1.1485	1.2233	1.2289	0.9702	0.9735	0.9819	0.9800	0.9730	**0.9105**
		DRP-2	1.1773	1.2210	1.2219	0.9787	0.9797	0.9949	0.9916	0.9763	**0.9102**
		P2D	2.4993	2.4501	1.3367	1.3028	1.2036	1.1943	0.8063	0.8079	**0.8022**
		SNILP	3.1473	3.1417	1.2073	1.1982	0.9066	0.9005	0.7716	0.7674	**0.7560**

**Remarks:** (*i*) background colors (from blue, through light blue, light red to red) of Table cells represent range (respectively, from the best to the worst one) of the model results based on the IQR(${\tilde{I}}_{flat}$) values obtained for each camera-*s* value combination; (*ii*) for each camera-model combination the values written with bold text represents the best approximation result obtained for the each of these combinations according to the IQR(${\tilde{I}}_{flat}$) measure.

**Table 5 sensors-23-01157-t005:** Comparison of the obtained η values for the DRP-1 and DRP-2 models.

Camera	Model	Degree of Approximation Polynomials *s*								
		2	3	4	5	6	7	8	9	10
**Cam-A**	DRP-1	1.0746	1.0775	1.0806	1.0832	1.0832	1.0834	1.0831	1.0832	**1.0824**
	DRP-2	1.0812	1.0859	1.0876	1.0892	1.0893	1.0895	1.0905	1.0905	**1.0901**
**Cam-B**	DRP-1	1.1473	1.1473	1.1511	1.1474	1.1490	1.1488	1.1512	1.1512	**1.1542**
	DRP-2	1.1469	1.1470	1.1509	1.1477	1.1507	1.1505	1.1527	1.1527	**1.1559**
**Cam-C**	DRP-1	1.3260	1.3220	1.2990	1.2903	1.2927	1.2964	1.2962	1.2946	**1.2936**
	DRP-2	1.3260	1.3214	1.2983	1.2899	1.2930	1.2976	1.2977	1.2962	**1.2952**
**Cam-D**	DRP-1	1.0602	1.0563	1.0541	1.0509	1.0500	1.0493	1.0495	**1.0500**	1.0499
	DRP-2	1.0692	1.0649	1.0626	1.0609	1.0594	1.0596	1.0598	**1.0605**	1.0606
**Cam-E**	DRP-1	1.0209	1.0162	1.0119	1.0078	1.0079	1.0086	1.0086	1.0078	**1.0061**
	DRP-2	1.0216	1.0169	1.0126	1.0086	1.0087	1.0094	1.0093	1.0085	**1.0068**

**Remarks:** (*i*) in each row the value written with bold text indicate value of η obtained for the best approximation result according to the STD(${\tilde{I}}_{flat}$) measure for a given camera-model pair; (*ii*) values with a gray background show the $\tilde{\eta }$ value of the individual camera.

## Data Availability

The data presented in this study are available on reasonable request from the corresponding author.

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
