# Peer review of "Image Vignetting Correction Using a Deformable Radial Polynomial Model"

_sensors, 2023, doi:10.3390/s23031157_

Round 1

Reviewer 1 Report

The reviewer would like to thank the authors for this thoughtful manuscript. This work has good potential. The authors are requested to put in some additional efforts to improve the quality of this manuscript. 

Introduction 

The authors are requested to cite the following articles on remote sensing and indicate the importance of image vignetting correction for critical Earth observation applications like natural disaster monitoring and hydrological estimation due to snowmelt from mountainous terrains.

-Shugar et al, A massive rock and ice avalanche caused the 2021 disaster at Chamoli, Indian Himalaya, Science, 2021

-Muhuri et al., “Performance Assessment of Optical Satellite-Based Operational Snow Cover Monitoring Algorithms in Forested Landscapes”, IEEE JSTARS, 2021. 

-Tsai, Y.L.S., et al., 2019. Remote Sensing of Snow Cover Using Spaceborne SAR: A Review. Remote Sensing.

Please also explain how the proposed correction technique can be used for both optical and radar satellite images. Please note that the Introduction section should ideally not have subsections. 

View Angle Normalization 

The authors are requested to explain how the normalization of the images are performed. A general strategy is to perform a cosine correction. Such discussion should be included in the manuscript.

Application in Remote Sensing

The authors have worked with an experimental setup where the source of illumination is close to the screen. This naturally creates the vignetting effect. In the context of remote sensing sensors the authors are requested to explain how the issue of vignetting seeps into the images and how the proposed technique can be still useful.

Tables 

The authors have presented a lot of statistics in the tables. Is it possible to provide a visual representation to only the important stats for the ease of comprehension. This way the article will have a better understanding and citation rate in the future. Furthermore, Table 5 is part of the methodology and not the results. The authors should clearly separate the sections into Introduction, Methodology & Data, Results, Discussion, and Conclusion.

Comparison of the Vignetting Estimates

These figures look more or less the same and the differences can’t be distinguished easily except for the boundaries. The authors can minimize the number of figures by only providing the comparison of vignetting corrections. Furthermore, it is better to provide a limited number of figures and show the contour plots separately since they are obscured by the overlying surface.

Discussion and Conclusion 

The authors are requested to list the key contributions in these sections. At the moment the sections are not detailed enough. Please have separate sections for Discussion and Conclusion. Discussion should not be included as a part of the Results section.

Reviewer 2 Report

  Review Comments

The authors have presented a new model of vignetting, i.e., the Deformable Radial Polynomial (DRP) model, which joins the simplicity of the RP model with the universality of more sophisticated models. The DRP model uses a simple distance transformation and minimization method to match the radial vignetting model to the non-radial vignetting of the analyzed lens-camera system. However, the following corrections can be considered by the authors to further improve the quality of the manuscript.

 I have some minor corrections and suggestions below:-

1. The abstract can be improved and the outcome of the work in terms of achieved performance calculations must be included in the abstract.

2. What novelty is established in this work compared to existing works? The novelty of the work can be highlighted better. Authors must show explain the novel contribution of the work with proper justification of the outcomes and also add at the end of the introduction.

3. Literature survey is missing. Authors must add a literature survey section proposed discussion of various papers published in proposed areas.

4. Comparative analysis of various performance parameters with respect to camera parameters must be discussed.

5. Comparative analysis of discussed parameters with respect to state of art methods is missing.

6. In Tables 3 and 4 presented models must be cited with specific citations.

7. The computational complexity of the algorithm must be discussed. Also, compare the proposed method in terms of computational complexity.

8. Future work and Limitations of the proposed work can be added and discussed.

Round 2

Reviewer 2 Report

All my concerns  and comments has been added and modified satisfactory. I Accept it in current form